# The Utility and Diagnostic Accuracy of Transient Elastography in Adults with Morbid Obesity: A Prospective Study

**DOI:** 10.3390/jcm11051201

**Published:** 2022-02-23

**Authors:** Ahmad Hassan Ali, Alhareth Al Juboori, Gregory F. Petroski, Alberto A. Diaz-Arias, Majid M. Syed-Abdul, Andrew A. Wheeler, Rama R. Ganga, James B. Pitt, Nicole M. Spencer, Ghassan M. Hammoud, R. Scott Rector, Elizabeth J. Parks, Jamal A. Ibdah

**Affiliations:** 1Division of Gastroenterology and Hepatology, University of Missouri, Columbia, MO 65211, USA; aliah@health.missouri.edu (A.H.A.); aljubooria@health.missouri.edu (A.A.J.); hammoudg@health.missouri.edu (G.M.H.); rectors@health.missouri.edu (R.S.R.); parksej@missouri.edu (E.J.P.); 2Biostatistics and Research Design Unit, School of Medicine, University of Missouri, Columbia, MO 65211, USA; petroskig@health.missouri.edu; 3Boyce & Bynum Pathology Professional Services, Columbia, MO 65201, USA; adiazarias@bbplpath.com; 4Department of Nutrition and Exercise Physiology, School of Medicine, University of Missouri, Columbia, MO 65211, USA; ms9rf@mail.missouri.edu; 5Department of Surgery, University of Missouri, Columbia, MO 65211, USA; wheeleraa@health.missouri.edu (A.A.W.); gangar@health.missouri.edu (R.R.G.); jpitt@columbiasurgical.com (J.B.P.); spencernm@columbiasurgical.com (N.M.S.); 6Research Service, Harry S Truman Memorial Veterans Medical Center, Columbia, MO 65201, USA; 7Department of Medical Pharmacology and Physiology, School of Medicine, University of Missouri, Columbia, MO 65211, USA

**Keywords:** fatty liver, morbid obesity, liver fibrosis, elastography, liver biochemistry

## Abstract

Patients with morbid obesity are at high risk for nonalcoholic fatty liver disease (NAFLD) complicated by liver fibrosis. The clinical utility of transient elastography (TE) by Fibroscan in patients with morbid obesity (body mass index (BMI) ≥ 40 kg/m^2^) is not well-defined. We examined the diagnostic accuracy of Fibroscan in predicting significant liver fibrosis (fibrosis stage ≥2) in morbidly obese patients (BMI ≥ 40 kg/m^2^). Patients scheduled for bariatric surgery were prospectively enrolled. Intraoperative liver biopsy, liver-stiffness measurement (LSM) by Fibroscan (XL probe), and biochemical evaluation were all performed on the same day. The endpoint was significant liver fibrosis defined as fibrosis stage ≥2 based on the Nonalcoholic Steatohepatitis Clinical Research Network. The optimal LSM cutoff value for detecting significant fibrosis was determined by using the Youden Index method. Routine clinical, laboratory, and elastography data were analyzed by stepwise logistic regression analysis to identify predictors of significant liver fibrosis and build a predictive model. An optimal cutoff point of the new model’s regression formula for predicting significant fibrosis was determined by using the Youden index method. One hundred sixty-seven patients (mean age, 46.4 years) were included, of whom 83.2% were female. Histological assessment revealed the prevalence of steatohepatitis and significant fibrosis of 40.7% and 11.4%, respectively. The median LSM was found to be significantly higher in the significant fibrosis group compared to those in the no or non-significant fibrosis group (18.2 vs. 7.7 kPa, respectively; *p* = 0.0004). The optimal LSM cutoff for predicting significant fibrosis was 12.8 kPa, with an accuracy of 71.3%, sensitivity of 73.7%, specificity of 70.9%, positive predictive value of 24.6%, negative predictive value of 95.5%, and ROC area of 0.723 (95% CI: 0.62–0.83). Logistic regression analysis identified three independent predictors of significant fibrosis: LSM, hemoglobin A1c, and alkaline phosphatase. A risk score was developed by using these three variables. At an optimal cutoff value of the regression formula, the risk score had an accuracy of 79.6% for predicting significant fibrosis, sensitivity of 89.5%, specificity of 78.4%, positive predictive value of 34.7%, negative predictive value of 98.3%, and ROC area of 0.855 (95% CI: 0.76–0.95). Fibroscan utility in predicting significant liver fibrosis in morbidly obese subjects is limited with accuracy of 71.3%. A model incorporating hemoglobin A1c and alkaline phosphatase with LSM improves accuracy in detecting significant fibrosis in this patient population.

## 1. Introduction

Nonalcoholic fatty liver disease (NAFLD) is currently the leading cause of chronic liver disease worldwide, with a reported global prevalence of 25.3% among adults ≥18 years old [1]. Furthermore, the rapidly increasing proportion of patients on the liver transplant waitlist or receiving liver transplant for nonalcoholic steatohepatitis (NASH) is alarming, rendering NASH among the top leading indications for liver transplantation [2], and the fastest growing cause of hepatocellular carcinoma in patients listed for liver transplantation [3].

By the year 2030, it is projected that one in every two adults will have obesity, and one in every four will have severe obesity (body mass index (BMI) ≥ 35 kg/m^2^) [4]. A close association between NAFLD and obesity has been firmly established. With NAFLD occurring frequently in people with morbid obesity, and in 80–90% of patients undergoing bariatric surgery [5,6], an estimated 20–47% of these patients have NASH, of which 8–12% progress to cirrhosis [7,8,9]. Thus, the assessment of liver fibrosis in patients with obesity undergoing bariatric surgery is crucial for risk stratification and making better-informed therapeutic decisions.

As in all other forms of chronic liver disease, liver biopsy is the gold standard for histopathological assessment of disease severity and fibrosis staging in patients with NAFLD. However, the use of liver biopsy is limited and is falling out of favor in clinical practice, due to procedure-associated pain, sampling error, inter- and intra-observer variation in interpretation, and the rare but fatal complications. Transient elastography (TE), a noninvasive tool for assessment of liver fibrosis, has been gaining increasing popularity and is recommended by leading societies [10,11]. Advantages of TE include its simplicity, short time of performance, painless nature, reproducibility, broader assessment of the hepatic parenchyma, and convenience of use in an outpatient setting. Its clinical utility and accuracy in discriminating between non-advanced and advanced fibrosis has been established in NAFLD, viral hepatitis and cholestatic liver disease. However, several drawbacks need to be acknowledged. TE is not an imaging technique, and therefore it is not possible to place the TE probe with certainty in an area of the liver parenchyma free of liver lesions and blood vessels [12]. Further, several patient-related factors, such as obesity, acute liver failure, elevated bilirubin, biliary obstruction, high degree of hepatic steatosis, infiltration disorders (e.g., amyloidosis), alcohol use, right-sided heart failure, and longer distance between the skin and liver capsule, have been shown to influence liver-stiffness measurement (LSM) by TE [12,13], and, therefore, caution should be exercised when interpreting LSM in these situations.

Early studies reported concerns regarding accuracy of the M probe when performing TE in patients with obesity, citing BMI and increased skin-to-liver capsule distance as risk factors for TE discordance with liver biopsy, unreliability, increased failure rates, and overestimation of fibrosis [14]. In response to these reports, the XL probe was developed, which significantly increased the accuracy and reliability of TE in patients with obesity. One study in patients with obesity reported a TE failure rate of only 1.1%, using the XL probe, compared to 16% failure rate using the M probe [15]. Later, several studies examined the utility of TE in patients with obesity undergoing bariatric surgery and reported high accuracy of TE in discriminating between non-advanced and advanced fibrosis comparable to that of the gold standard liver biopsy [16]. However, there are factors other than BMI that can influence the LSM in patients with obesity. A recent study reported that the SCD was found to have a more profound effect on the accuracy of LSM results than the BMI in patients with obesity, suggesting that SCD should be taken into consideration when assessing the liver stiffness in this patient population [17]. Although these studies included patients with obesity, the range of BMI was wide, and studies examining the utility of TE exclusively in patients with morbid obesity (BMI ≥ 40 kg/m^2^) are lacking.

The present prospective investigation was undertaken to include only patients with BMI ≥ 40 kg/m^2^, who underwent bariatric surgery at our institution, with the following goals: (a) examine the diagnostic accuracy and reliability of Fibroscan as means for detecting significant fibrosis in morbidly obese subjects, (b) identify an optimal cutoff value for detecting significant fibrosis in morbidly obese subjects, and (c) build a model for predicting significant fibrosis in patients with BMI ≥ 40 kg/m^2^. We hypothesize that the diagnostic performance of Fibroscan is affected in patients with morbid obesity.

## 2. Materials and Methods

### 2.1. Institutional Review Board Approval and Subjects’ Consenting

This study was approved by the University of Missouri Institutional Review Board (#2008258). Study subjects were enrolled between July 2017 and November 2019 through the Bariatric clinic at the University of Missouri School of Medicine, Columbia, MO, USA. All study subjects provided written informed consent, and all procedures were performed in accordance with ethical standards of the institution’s bylaws and research policies and with the 1964 Declaration of Helsinki and its later amendments.

### 2.2. Study Subjects

Study subjects were invited and recruited consecutively once the date of bariatric surgery was determined. Subjects were excluded if they had a history of alcohol intake >20 g per day or liver disease based on history and/or laboratory data. Patients with BMI ≥ 40 kg/m^2^ were included in this study. On the day of surgery, subjects reported to the surgical suite in the fasted state. Two hours before surgery, blood samples were collected before anesthesia for measurement of lipid profile, complete blood count, blood glucose, hemoglobin A1c (HbA1c), and liver chemistry (alkaline phosphatase (ALP), aspartate and alanine aminotransferase (AST and ALT), and albumin) at a CLIA-certified laboratory.

### 2.3. Liver-Stiffness Measurement by Transient Elastography

One hour prior to the scheduled bariatric surgery, LSM was performed on each subject with a Fibroscan^®^ (FibroScan Compact 530, Echosens, Paris, France), using an XL probe in accordance with the manufacturer’s instructions. Two certified study investigators performed the LSM. Each study subject laid in the supine position, with his/her head and legs directed toward the opposite side of the Fibroscan to increase the intercostal space for optimal measurement of LSM and controlled attenuation parameter (CAP). The study investigators paid attention to the breathing pattern of each study subject so that the measurements were consistent and performed at the same point. The Fibroscan XL probe was placed perpendicular to the body at the point where two theoretical lines running from the xiphoid process and the mid-axillary line meet. At least 12 LSM readings with interquartile range (IQR)-to-median (M) ratio (IQR/M) of ≤30% was considered an accurate LSM and CAP reading. The success rate was calculated as the number of successful measurements divided by the total number of measurements made. A Fibroscan result was considered reliable if it met all the following criteria: number of valid measurements ≥10, success rate ≥60%, and IQR/M <30%. LSM values were expressed as kilopascals (kPa), and CAP values were expressed as dB/m. The investigators were blinded to the histological stage data at the time of performing Fibroscan by nature of the study, since the Fibroscans were performed prior to bariatric surgery.

### 2.4. Liver Tissue Sampling

To minimize the potential risk of liver injury and the resultant influx of inflammatory cells caused by anesthesia and/or liver manipulation, liver tissue was obtained after initiation of anesthesia according to standardized protocols via a wedge biopsy of the left lobe of the liver, using either an ultrasonic dissector or bipolar or monopolar cautery. The specimen was extracted through a 12 mm trocar. Hemostasis was achieved by electrocautery.

### 2.5. Liver Biopsy Interpretation

Liver tissue was placed in 10% formalin. Liver-wedge specimens were bisected and submitted entirely in one block. Hematoxylin and Eosin stain and Masson’s trichrome stain were performed on specimens. All specimens were interpreted, graded, and staged by one experienced liver pathologist who was blinded to the clinical, laboratory, and elastography data. NASH grading and fibrosis staging were performed by using the pathological grading and staging system proposed by the NASH Clinical Research Network [18]. The diagnosis of steatohepatitis required the presence of all the three following histopathological components: steatosis, lobular inflammation, and hepatocellular ballooning [19]. Briefly, steatosis was graded based on the percentage of hepatocytes involved: 0 for <5%, 1 for 5–33%, 2 for >33–66%, and 3 for >66% [18]. Identification of lobular inflammation required the presence of intra-acinar and portal inflammation characterized by inflammatory foci of polymorphonuclear leukocytes, lymphocytes, eosinophils, and microgranulomas and was graded 0–3 based on the inflammatory foci per 200× field: 0 for none, 1 for <2 foci, 2 for 2–4 foci, and 3 for >4 foci [20]. Hepatocellular ballooning, defined as swollen-appearing hepatocytes indicative of severe cell injury, was evaluated for zonal location, and the severity estimation was based on the numbers of hepatocytes affected: 0 for none, 1 for few, and 2 for many [21]. Finally, fibrosis was staged as follows: stage 0, no fibrosis; stage 1, perisinusoidal or periportal fibrosis; stage 2, perisinusoidal and portal/periportal fibrosis; stage 3, bridging fibrosis; and stage 4, cirrhosis [18]. Significant fibrosis was defined as fibrosis stage ≥2, whereas advanced fibrosis was defined as fibrosis stage ≥3. 

### 2.6. Statistical Analysis

Continuous variables are expressed as median with minimum and maximum, and categorical variables are expressed as frequency and percentage. Study subjects were categorized according to the presence or absence of significant fibrosis (F0–F1 vs. F2–F4). Fibrosis stage 2 as the cutpoint for significant fibrosis was chosen because of the high risk of liver-related death in NAFLD/NASH patients with fibrosis stage ≥2 reported by a recent meta-analysis study [22]. For each continuous candidate predictor, we evaluated normality with histograms and examined Shapiro–Wilk and Kolmogorov–Smirnov tests of normality [23]. Baseline clinical, laboratory, and elastography variables were compared between the two groups by using the Chi-squared test for categorical variables and the Wilcoxon Rank Sum test for the continuous variables. Multicollinearity was assessed by using the variance inflation factor (VIF) for each predictor. The VIF for each variable is a simple function of the R^2^ for regressing that variable on the other predictors. A predictor that is uncorrelated with all other predictors will have a VIF of 1.0. Rules of thumb suggest that a VIF > 10 indicates deleterious multicollinearity while others suggest a VIF of 5.0 is cause for concern [24]. The optimal cutoff LSM value for detecting significant fibrosis (i.e., F ≥ 2) was based on the Youden Index as estimated by the Empirical Cutpoint Estimation available in Stata [25], and the identified optimal estimated cutoff LSM was adjusted according to the method of Fluss et al. [26]. The sensitivity, specificity, positive predictive value, negative predictive value, and the area under ROC at the optimal cutpoint were calculated.

An important aspect of a predictive model is its ability to generalize to an independent new dataset. Evaluating the predictive performance of a model (in this case, the LSM) by using all the cases from the original sample tends to result in an overly optimistic estimate of the predictive model. K-fold cross-validation follows the concept of splitting the dataset into training and test datasets. Using K-fold cross-validation, we iterated over a dataset k times. In each round, the data were split into *k* parts: one part used as the test dataset, and the remaining *k*-1 used as the training dataset. The cross-validated AUROC was then calculated by averaging the AUCs corresponding to the *k* parts and the bootstrap procedure was applied to the cross-validated AUC to obtain statistical inference and 95% bias corrected confidence intervals (CIs). The K-fold cross-validation procedure and constructing the cross-validated ROC curves were performed by using the CVAUROC Stata module [27]. 

#### Developing a Model for Predicting Significant Fibrosis 

Univariate descriptive statistics were used to compare patients with and without significant fibrosis. Variables with *p*-value < 0.05 on univariate analysis were included in logistic regression analysis. Variables that were consistently selected in stepwise selection, forward selection, and backward elimination logistic regression were included in the final model. For stepwise and forward selection, a *p*-value < 0.05 was used as the criterion for entry into the predictive model, whereas, for backward elimination, a *p*-value ≥ 0.05 was used as the criteria for exiting the model.

If *p* is the probability of significant fibrosis, then the logistic regression model is defined by the following formula: *Log(p*/1 *− p) = β*_0_
*+ β*_1_*X*_1_
*+ β*_2_*X*_2_
*+ β*_3_*X*_3_, where *βs* are the coefficients and the *Xs* are candidate predictors. Using the Youden Index method, the optimal cutpoint for the linear predictor values was calculated to identify the presence of significant fibrosis. Calibration of the final new model was assessed with the Hosmer–Lemeshow goodness-of-fit test [28] and by fitting a smooth calibration curve to the predicted probabilities and comparing that curve to the reference line [29]. K-fold cross-validation was used to test the ability of the final model to predict significant fibrosis in a training dataset. The diagnostic performance parameters of the new model (sensitivity, specificity, positive predictive value, negative predictive value, accuracy, and the ROC area) were calculated and compared to the models developed by Kao et al. [30] and Newsome et al. [31], both of which were specifically developed to predict significant liver fibrosis, i.e., F ≥ 2, in patients with NAFLD/NASH. The Kao risk model combines LSM and aspartate aminotransferase/platelet ratio index (APRI) and is calculated by the weighted sum of the β-coefficients of LSM (2 points for LSM > 7 kPa) and APRI (1 point for APRI > 0.4). The range of this score is 0 to 3, with 1 being the cutoff point for significant liver fibrosis. The Fibroscan-AST (FAST) score incorporates LSM and AST and is calculated by using a formula [31]. We then compared the ROC area of our final new model to that of the Kao and FAST score, using a nonparametric test introduced by DeLong et al. [32]. Statistical analyses were conducted by using STATA version 12.1 (StataCrop LP, College Station, TX, USA). The calibration plot of the observed-against expected probabilities for assessment of the new prediction model’s performance was constructed by using the pmcalplot package in Stata [33]. We used the roccomp test package in Stata [34] to compare between the ROC area of our final new model to that of the Kao score and the FAST score. 

## 3. Results

### 3.1. Subject Characteristics 

A total of 235 patients consented to undergo bariatric surgery at our institution between July 2017 and November 2019. Of the 235 subjects, 68 were excluded from this study (Figure 1), and the remaining 167 enrolled subjects had complete clinical phenotyping, laboratory, histological, and transient elastography data available for analysis. 

Their mean age at the time of bariatric surgery was 46.4 years (range: 22.9–77.2 years), and the majority (83.2%) was female. Histologically, the prevalence of steatosis alone was 29.3% (49/167). Furthermore, the prevalence of steatohepatitis (with or without fibrosis) was 40.7% (68/167). Lastly, the prevalence of F ≥ 2, F3, and F4 was 11.4%, 5.4%, and 1.8%, respectively. Additional clinical and laboratory data are shown in Table 1 and Table 2, respectively.

A comparison between the two groups, F0–F1 vs. F2–F4, is presented in Table 1, Table 2 and Table 3. Upon univariate analysis, the frequency of type 2 diabetes was significantly higher in those with significant fibrosis compared to the no or non-significant fibrosis group (68.4% vs. 26.0%, respectively; *p* < 0.0001). Subjects with significant fibrosis had significantly higher levels of glucose, HbA1c, ALP, AST, ALT, and TG compared to those with no or non-significant fibrosis (Table 1 and Table 2).

### 3.2. Fibroscan Measurements

Ten or more valid measurements were obtained in 99.4% (166/167) of the study cohort, with a mean overall success rate of 58.8% (i.e., on average, less than two acquisitions were needed for every one valid Fibroscan measurement). Furthermore, 55.7% of the subjects had a recorded Fibroscan success rate over 60%. A reliable Fibroscan (defined as success rate ≥ 60%, IQR/M ≤ 0.3, and valid measurements ≥ 10) was recorded in 52.7% (88/167) of the entire cohort. There was no statistically significant difference in the number of valid Fibroscan measurements, success rate, or in the reliability between the two groups (F0–F1 vs. F2–F4, Table 3). Importantly, LSM and CAP were significantly higher in the significant fibrosis group compared to the no or non-significant fibrosis group (median LSM = 18.2 vs. 7.7 kPa, *p* = 0.0004; and the median CAP = 361.4 vs. 317.9 dB/m, *p* = 0.002, respectively). Figure 2 shows a boxplot distribution of LSM values over fibrosis stages.

### 3.3. Diagnostic Performance and Cross Validation of LSM in Discriminating between Presence or Absence of F ≥ 2

Using the Youden Index method, we found the optimal cutoff value of LSM-identifying subjects with F ≥ 2 to be 12.8 kPa, with a sensitivity of 73.7%, specificity of 70.9%, positive predictive value of 24.6%, negative predictive value of 95.5%, and an accuracy of 71.3%. The ROC area was 0.723, with a 95% CI of 0.62 to 0.83.

Using k = 5, we split the original sample (*n* = 167) into five subsamples: two subsamples each with *n* = 34, and three subsamples each with *n* = 33. During each round of cross-validation, one subsample was used as the test dataset, and the remaining four were used as the training dataset. This process was repeated five times so that each subsample was used as a test dataset. Finally, the cross-validated mean ROC area of the five fitted logistic regression model estimates was 0.746, with a bootstrap bias corrected 95% CI of 0.43 to 0.81.

### 3.4. Clinical Phenotyping of those with False-Positive Fibroscan Results 

Of those with F0–F1, 29.1% (43/148) had LSM values ≥ our proposed cutoff for F ≥ 2 (12.8 kPa) and were classified as false positive, whereas 70.9% had LSM < our proposed cutoff for F ≥ 2 (12.8 kPa), and those were classified as true negative (Table 4). Subjects classified as false positives had significantly higher median LSM values compared to the true negatives (19.6 vs. 6.7 kPa, *p* < 0.0001). This finding led us to hypothesize that the group of subjects with false positive Fibroscan results (i.e., F0–F1 and LSM ≥ 12.8 kPa) are clinically different than those classified as true negatives (i.e., F0–F1 and LSM < 12.8 kPa). Indeed, the false-positive group had significantly higher BMI and CAP values compared to the true negative group. Further, the frequency of BMI > 45, >50, >55, and >65 was significantly higher in the false-positive group compared to the true-negative group (Table 5).

### 3.5. Predictive Model Building

Prior to model building, we evaluated the multicollinearity of all candidate predictors. Only ALT and AST had VIFs of slightly more than 5.0. Upon stepwise selection, forward selection, and backward elimination logistic regression, LSM, HbA1c, and ALP were consistently selected. In the final three-variable model consisting of LSM, A1c, and ALP, the largest VIF was associated with A1c with a value of 1.07. The collinearity between AST and ALT may have prohibited them from being selected into the final new model. However, forcing them into the three-variable new model did not meaningfully improve the discrimination. The final model consisted of the three variables: LSM (odds ratio (OR) = 1.06; 95% CI, 1.02 to 1.10; β = 0.059; *p* = 0.002; 95% CI, 0.02 to 0.09); HbA1c (OR = 1.59; 95% CI, 1.16 to 2.17; β = 0.461; *p* = 0.004; 95% CI, 0.15 to 0.78); and ALP (OR = 1.04; 95% CI, 1.01 to 1.06; β = 0.036; *p* = 0.005; 95% CI, 0.01 to 0.06). The cross-validated mean ROC area of the five fitted logistic regression model estimates of the three variables was 0.867, with a bootstrap bias corrected 95% CI of 0.67 to 0.95 (Figure 3).

We looked for any potential interaction between the three predictors; no significant interaction was noted. The model calibration was adequate, as reflected in a non-significant Hosmer–Lemeshow goodness-of-fit test (*p* = 0.567), and as illustrated in a calibration plot with 95% confidence bands for the calibration curve with the reference line over the full range of estimated probabilities included (Figure 4). 

The final regression formula (risk score) for predicting significant liver fibrosis based on the three variables, the *Fib*rosis *R*isk score in the morbidly *O*bese-3 (FibRO-3), is as follows:FibRO-3 = 0.059 × LSM (kPa) + 0.461 × HbA1c (%) + 0.036 × ALP (U/L).

Using the Youden method, we determined that the optimal cutoff point for the regression formula was 6.603; the summary statistics for the linear prediction values were shown in Figure 5. At this FibRO-3 cutoff point, the sensitivity for predicting significant fibrosis was 89.5%, specificity was 78.4%, positive predictive value was 34.7%, negative predictive value was 98.3%, accuracy was 79.6%, and the ROC was 0.855 (95% CI, 0.76 to 0.95). 

The performance of the new predictive model, the FibRO-3, was compared to that reported by Kao et al. [30] and the FAST score [31]. The ROC area of the FibRO-3 model (0.855; 95% CI, 0.76 to 0.95) was superior to that of the Kao score (0.579; 95% CI, 0.49 to 0.67) and the FAST score (0.708; 95% CI, 0.59 to 0.83). Furthermore, the FibRO-3 had significantly higher sensitivity compared to the FAST score (89.5% vs. 57.9%), but a lower specificity (78.4% vs. 83.8%). The percentage of correctly classified patients was similar between the FibRO-3 and the FAST score (79.6% vs. 80.8%). The Kao score had the lowest specificity (31.8%) and accuracy (37.7%) among the three predictive models. Finally, the ROC area for our new FibRO-3 model performed statistically significantly better than the FAST score (*p* = 0.009) and the Kao et al. score (*p* = <0.0001).

## 4. Discussion

### 4.1. Main Findings

In view of the alarmingly increasing incidence and prevalence of obesity and its liver-related morbidity, accurate and reliable methods of assessing severity of liver disease in terms of fibrosis staging are desperately needed. Undoubtedly, LSM using TE is one of the most accurate and validated tools for evaluation of patients with chronic liver disease. In patients with morbid obesity complicated by NAFLD/NASH, TE has been shown to be a reliable method for quantifying liver fibrosis [16]. However, the previous studies included patients with a wide range of BMI. The present study is the first in the USA to assess the diagnostic accuracy of LSM by TE exclusively in patients with morbid obesity (BMI ≥ 40 kg/m^2^). This study has several features to highlight. In a well-defined cohort of patients with morbid obesity undergoing bariatric surgery, at least ten valid Fibroscan measurements were possible in all patients but one (99%). Furthermore, a reliable Fibroscan was accomplished in more than one-half of the patients. These Fibroscan reliability parameters are better than those reported by Weiss et al. [35], who reported a reliability of 41%, and at least 10 valid Fibroscan measurements in only 22% of their cohort with morbid obesity. Most important, the diagnostic performance of Fibroscan was adequate in identifying those with fibrosis stage ≥2, but significantly improved when the independent predictors HbA1c and ALP were added to a predictive model (FibRO-3) combining the three variables, namely LSM, HbA1c, and ALP. Finally, our predictive model (FibRO-3) outperformed the Kao et al. score [30] and the FAST score [31] in terms of ROC area, had a higher sensitivity for detecting fibrosis stage ≥2 compared to the FAST score, and had an accuracy comparable to that of the FAST score.

### 4.2. The Utility of Fibroscan in Patients with Morbid Obesity

The accuracy of Fibroscan in patients with NAFLD and obesity is well-established; however, data regarding its use in patients with morbid obesity are very limited. As far as we know, only one study conducted in Germany examined the use of Fibroscan in patients with BMI ≥ 40 kg/m^2^ [35]. However, that study included a smaller number of patients (*n* = 87) and did not report an optimal LSM threshold for diagnosing liver fibrosis. When we applied the Castera et al. threshold rules to our cohort, using an LSM cutoff of 7.1 kPa for fibrosis stage ≥2 [36], the specificity, accuracy, and ROC area for identifying fibrosis stage ≥2 were 43.2%, 47.3%, and 0.611, respectively. When equal weight was given to the sensitivity and specificity, using the Youden method, the optimal LSM cutoff for identifying fibrosis stage ≥2 in the present cohort was 12.8 kPa. At this LSM cutoff, the specificity, accuracy, and ROC area were 70.9%, 71.3%, and 0.723, respectively. Obviously, our LSM cutoff value for detecting fibrosis stage ≥2 is higher than the standard LSM thresholds reported in the literature (ranging between 6.6 and 7.8 kPa) [37], which intrigued us to further explore this discrepancy. To do so, we compared the clinical features of the false-positive cases (i.e., those with F0–F1 and LSM ≥ 12.8 kPa) to the true-negative cases (i.e., those with F0–F1 and LSM < 12.8 kPa). Interestingly, those in the false-positive group had significantly higher BMI and CAP values. Furthermore, the frequency of those with BMI > 45, >50, >55, and >65 was significantly higher in the false-positive compared to the true-negative group. Thus, LSM was higher in the false-positive group likely due to the higher BMI and CAP compared to the true-negative group. Furthermore, although not measured, we speculate that longer skin-to-liver capsule distance (SCD) could have contributed to the higher median LSM values in the false-positive group. Indeed, one study reported higher median LSM in patients without advanced liver fibrosis with SCD longer than 35 mm, thus overestimating the LSM in patients with morbid obesity [38]. Our data add to the existing literature, which suggest that higher BMI and CAP values result in falsely elevated LSM values [39,40]. These data emphasize the need for larger studies to examine the accuracy of the current standard LSM thresholds for discriminating fibrosis stage ≥2 in patients with morbid obesity.

### 4.3. Predictive Model: FibRO-3

It is intriguing that none of the traditionally known liver chemistries, particularly AST and ALT, was selected as a candidate variable in the final FibRO-3 model. In fact, adding one or both did not improve the performance of the model (data not shown). Interestingly, ALP was consistently selected as a prognosticator of fibrosis stage ≥2. In the previously published NASH scoring systems, study populations were recruited from liver clinics who had higher mean AST and ALT values (~50 and 60 U/L, respectively) compared to the mean values in the patient population in the present study (32 and 37 U/L, respectively) [41,42,43]. This difference in the mean values of hepatic transaminases is due to the inherent feature of the present population, as they were recruited from a bariatric clinic. As Ooi et al. pointed out, prognostic scores such APRI, FIB-4, and the NAFLD fibrosis scores were primarily developed in patients who were referred to hepatology practices for abnormal liver chemistry, and, hence, a higher prevalence of advanced liver disease (fibrosis stage ≥3 in ~40%) was reported in these studies [44]. Thus, AST and ALT in the morbidly obese patients may have less predictive value. The results of the present study, as well as those reported by Kao et al. [30] and Ooi et al. [44], call for establishing newer predictive models or modifying thresholds of the existing ones for more accurate risk-stratification in morbidly obese patients. 

### 4.4. Study Strengths

Our study has several strengths. In addition to its prospective nature, subjects were consecutively enrolled through a bariatric clinic, thus eliminating any potential selection bias. Further, it is the first USA study to examine the diagnostic performance of Fibroscan in a well-defined bariatric cohort exclusively with BMI ≥ 40 kg/m^2^. The FibRO-3 model combines a noninvasive widely available and validated tool for fibrosis assessment with routinely obtained laboratory tests. Moreover, clinical, laboratory, elastography, and histology data were all obtained on the same day of surgery. An experienced liver pathologist who interpreted liver histology in the study subjects was blinded to the patients’ data. Only the XL probe of the Fibroscan was used in this study, thus eliminating the bias introduced by using the smaller size (M) probe that has been shown to be associated with Fibroscan failure, defined as no valid measurements.

### 4.5. Study Limitations

Our study had some limitations. As is typical with bariatric populations, most patients in the present cohort were female. More than 90% were Caucasians, thus mirroring the racial makeup of the Midwest region of the USA. The lower prevalence of this study’s endpoint (fibrosis stage ≥2) is another drawback, but it is an inherent limitation and reflects the nature of the population under study who were recruited from an outpatient bariatric setting. Finally, the performance of our proposed predictive model needs to be validated by an external independent cohort.

## 5. Conclusions

In a bariatric patient population with BMI ≥ 40 kg/m^2^ in whom the prevalence of steatohepatitis and fibrosis stage ≥2 were 40.7% and 11.4%, respectively, LSM alone had a relatively good performance in identifying those with fibrosis stage ≥2 when an LSM cutoff value of 12.8 kPa was used. HbA1c and ALP were found to be independent predictors of fibrosis stage ≥2, and a model combining the three parameters (LSM, HbA1c, and ALP), the FibRO-3, had a better diagnostic performance in identifying fibrosis stage ≥2 compared to LSM alone. Based on these findings, it is prudent to re-examine the current standard LSM thresholds in identifying clinically important stages of fibrosis in morbidly obese patients. Our study suggests that higher-than-standard LSM cutoff values might be needed for more accurate risk-stratification of this patient population.

## Figures and Tables

**Figure 1 jcm-11-01201-f001:**
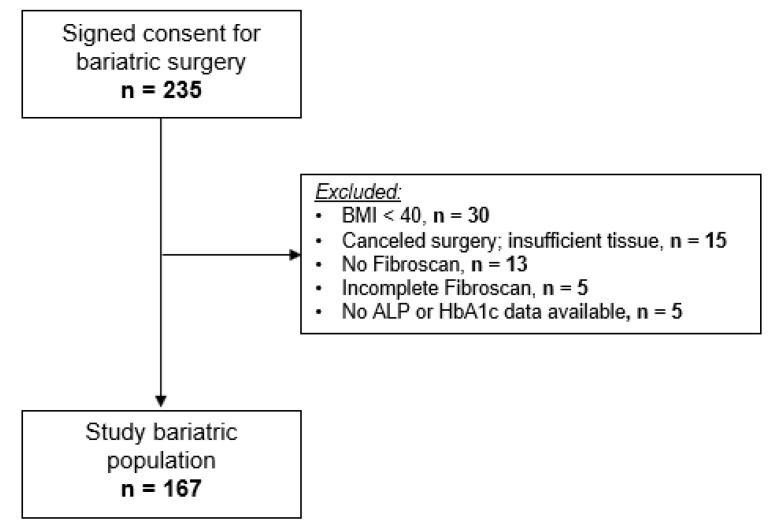
Flowchart of the patient population.

**Figure 2 jcm-11-01201-f002:**
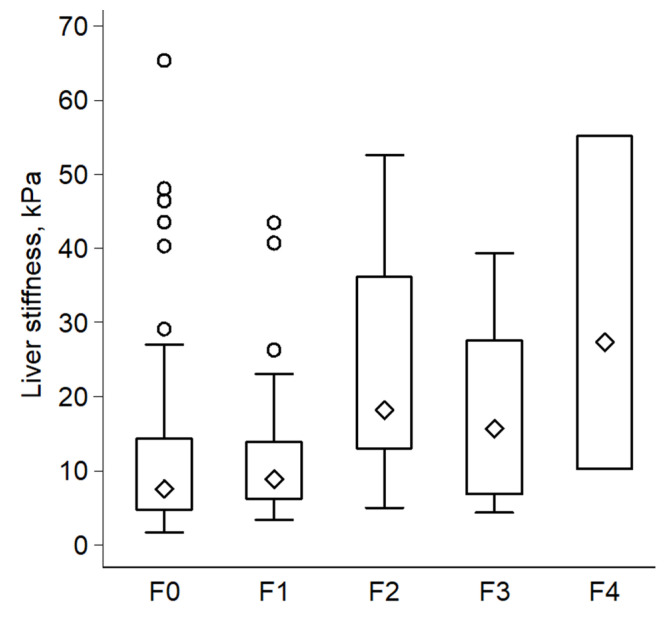
Boxplot illustrates distribution of liver-stiffness measurements over histological stages of fibrosis (*n* = 167).

**Figure 3 jcm-11-01201-f003:**
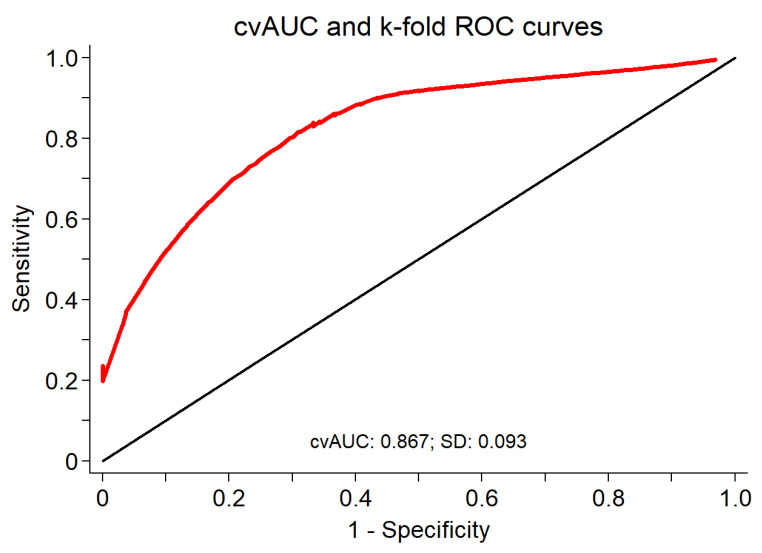
Cross-validated mean area under ROC for LSM, HbA1c, and ALP as a diagnostic test for significant fibrosis (*n* = 167).

**Figure 4 jcm-11-01201-f004:**
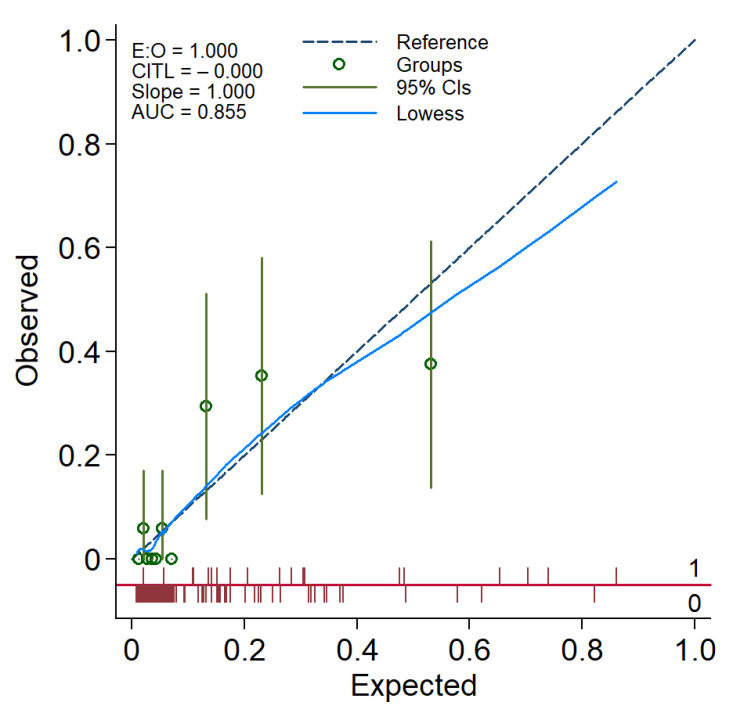
A calibration plot demonstrating the performance of the FibRO-3 model. The expected risk of significant liver fibrosis (stage ≥2) is divided into 10 equally sized groups (of tenths). The green circles and spikes on the diagonal line are average predicted risks and 95% confidence bands, respectively. The dotted straight line represents the reference line of the model’s calibration. The blue line connecting the green circles is the locally weighted scatterplot smoothing (LOWESS).

**Figure 5 jcm-11-01201-f005:**
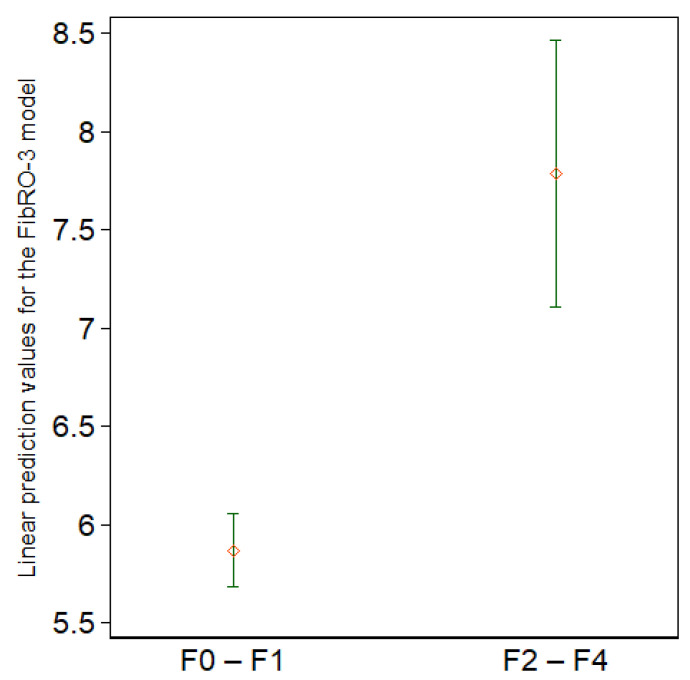
Summary statistics for the linear prediction values according to the presence of significant fibrosis (F0–F1 vs. F2–F4). The diamonds represent the means, whereas the upper and lower short horizontal lines represent the upper and lower ends of the 95% confidence intervals, respectively (*n* = 167).

**Table 1 jcm-11-01201-t001:** Demographic and clinical features of the enrollee subjects who underwent bariatric surgery at our institution between July 2017 and November 2019 (*n* = 167).

Variable	All Subjects	F0–F1	F2–F4	*p*-Value
	(*n* = 167)	(*n* = 148)	(*n* = 19)	
Age				
Median	46	45	53	0.037
Min; Max	22.9; 77.2	22.9; 77.2	29.0; 65.7	
Gender, female				
Percentage	83.2%	82.4%	89.5%	0.45
Frequency	139	122	17	
Tobacco use *, yes				
Percentage	48.5%	48.6%	47.4%	0.92
Frequency	80	71	9	
Type 2 diabetes *, yes				
Percentage	30.9%	26.03%	68.4%	<0.0001
Frequency	51	38	13	
Hypertension *, yes				
Percentage	53.3%	50.7%	73.7%	0.059
Frequency	88	74	14	
Hyperlipidemia *, yes				
Percentage	44.9%	41.5%	63.2%	0.09
Frequency	74	62	12	
Body weight, kg				
Median	131.9	131.2	143.4	0.52
Min; Max	93.4–238.9	93.4; 238.9	106.9; 173	
Body mass index, kg/m^2^				
Median	48	48	53	0.22
Min; Max	40; 67.3	40; 67.3	40; 63	

* Data are missing for two patients in the F0–F1 group.

**Table 2 jcm-11-01201-t002:** Laboratory features of the enrollee subjects who underwent bariatric surgery at our institution between July 2017 and November 2019 (*n* = 167).

Variable	All Subjects	F0–F1	F2–F4	*p*-Value
	(*n* = 167)	(*n* = 148)	(*n* = 19)	
Glucose, mg/dL				
Median	94	92	120	<0.0001
Min; Max	57; 211	57; 211	87; 181	
HbA1c, %				
Median	5.7	5.7	7.3	<0.0001
Min; Max	4.3; 13.2	4.3; 13.2	5.3; 10.5	
Albumin, g/dL				
Median	4.3	4.3	4.6	0.08
Min; Max	3.4; 5.4	3.4; 5.3	4.0; 5.4	
ALP, U/L				
Median	67	66	88	0.0004
Min; Max	26; 157	26; 157	53; 127	
AST, U/L				
Median	26	25	38	0.002
Min; Max	9; 152	9; 152	17; 125	
ALT, U/L				
Median	28	27	42	0.008
Min; Max	9; 273	9; 186	14; 273	
Hgb, g/dL				
Median	13.6	13.6	14.1	0.14
Min; Max	9.5; 16.7	9.5; 16.4	12.0–16.7	
Platelets, cells × 10^9^				
Median	261	260	283	0.98
Min; Max	117; 510	138; 510	117; 437	
TC, mg/dL				
Median	160	160	171	0.92
Min; Max	77; 265	104; 265	77; 244	
TG, mg/dL				
Median	119	117	150	0.032
Min; Max	48; 329	48; 329	78; 243	
LDL, mg/dL				
Median	96	97	96	0.56
Min; Max	21; 205	35; 205	21; 167	
HDL, mg/dL				
Median	39	39	37	0.93
Min; Max	20; 82	20; 82	27; 64	

Abbreviations: ALP, alkaline phosphatase; AST, aspartate aminotransferase; ALT, alanine aminotransferase; TC, total cholesterol; TG, triglycerides; LDL, low-density lipoprotein; and HDL, high-density lipoprotein.

**Table 3 jcm-11-01201-t003:** Transient elastography features of enrollee subjects who underwent bariatric surgery at our institution between July 2017 and November 2019 (*n* = 167).

Variable	All Subjects	F0–F1	F2–F4	*p*-Value
	(*n* = 167)	(*n* = 148)	(*n* = 19)	
LSM, kPa				
Median	8.3	7.7	18.2	
IQR	5–15.7	4.9–14.4	10.2–28	0.0004
Min; Max	1.7; 65.3	1.7; 65.3	4.3; 55.1	
IQR/M				
Median	0.19	0.19	0.19	0.52
Min; Max	0.06; 0.38	0.06; 0.38	0.09; 0.3	
Valid measurements, yes				
Median	14	14	14	0.76
Min; Max	9; 64	9; 64	12; 22	
≥10 valid measurements				
Percentage	99.4%	99.3%	100%	0.72
Frequency	166	147	19	
Success rate, %				
Median	58.8	58.1	63.4	0.48
Min; Max	0.95; 100.0	0.95; 100.0	6.7; 100.0	
Success rate ≥ 60%				
Percentage	55.7%	54.7%	63.2%	0.49
Frequency	93	81	23	
Reliable Fibroscan, yes				
Percentage	52.7%	52.03%	57.9%	0.63
Frequency	88	77	11	
CAP, dB/m				
Median	329	322	382	0.002
Min; Max	100; 400	100; 400	249; 400	

Abbreviations: LSM, liver-stiffness measurement; IQR, interquartile range; IQR/M, interquartile range/median; CAP, controlled attenuation parameter.

**Table 4 jcm-11-01201-t004:** Classification of the enrollee subjects who underwent bariatric surgery according to the presence of significant fibrosis (F ≥ 2) and LSM ≥ 12.8 kPa.

	Fibrosis ≥ 2		
LSM ≥ 12.8 kPa	Absent	Present	Total
No	True Negative	False negative	
	105	5	110
Yes	False positive	True positive	
	43	14	57
Total	148	19	167

Abbreviations: LSM, liver-stiffness measurement; kPa, kilopascals.

**Table 5 jcm-11-01201-t005:** Comparison between enrollee subjects with F0–F1 according to their LSM values (≥12.8 kPa false-positive cases and <12.8 kPa true-negative cases).

Variable	Group		*p*-Value
	False Positive (*n* = 43)	True Negative (*n* = 105)	
Age			
Median	48	44	0.19
Diabetes, yes			
Percentage	45.2%	18.3%	0.002
Frequency	(19/42)	(19/104)	
Weight, kg			
Median	141.2	129.1	0.007
BMI, kg/m^2^			
Median	52.0	44.6	0.0005
CAP, dB/m			
Median	371.0	310.0	0.0003
BMI > 45			
Percentage	86.1%	53.3%	<0.0001
Frequency	37	56	
BMI > 50			
Percentage	60.5%	29.5%	<0.0001
Frequency	26	31	
BMI >55			
Percentage	30.2%	14.3%	0.025
Frequency	13	15	
BMI > 60			
Percentage	16.3%	6.7%	0.07
Frequency	7	7	
BMI > 65			
Percentage	9.3%	0.95%	0.01
Frequency	4	4	
Fibrosis stage 0 (histology)			
Percentage	83.7%	80.0%	0.65
Frequency	36	84	
Fibrosis stage 1 (histology)			
Percentage	16.3%	20.0%	0.65
Frequency	7	21	
Severe steatosis (≥66%)			
Percentage	16.3%	18.1%	0.79
Frequency	7	19	

Abbreviations: BMI, body mass index; CAP, controlled attenuation parameter.

## Data Availability

The data presented in this study are available upon request from the corresponding author. The data are not publicly available in order to protect the privacy of the participating subjects, as outlined in our study protocol.

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
