# Peer review of "The Utility and Diagnostic Accuracy of Transient Elastography in Adults with Morbid Obesity: A Prospective Study"

_jcm, 2022, doi:10.3390/jcm11051201_

Round 1

Reviewer 1 Report

Ali and Juboori et al provide a highly predictive model in FibRO-3 to determine liver fibrosis in morbidly obese patients with NAFLD. The method uses available equipment (fibroscan) and routine bloods and performs well compared to previous studies. It is also highly useful given the high BMI range examined and the prevalence of diabetes and liver fibrosis. 

One minor comment, in tables 1, 2 and 3 the data presented nominally in 3 groups, to compare between these groups multivariate analysis should be performed.   

Reviewer 2 Report

In this manuscript entitled “The utility and diagnostic accuracy of transient elastography in adults with morbid obesity: A prospective study.” Ahmad Hassan Ali and his co-workers examined the diagnostic accuracy of Fibroscan in predicting significant liver fibrosis (fibrosis stage ≥ 2) in morbidly obese patients n=167 (BMI ≥ 40 kg/m2) scheduled for bariatric surgery with nonalcoholic fatty liver disease. The optimal LSM cutoff value for detecting significant fibrosis was determined using the Youden Index method. The optimal LSM cutoff for predicting significant fibrosis in this patient population was 12.8 kPa with accuracy of 71.3%. In addition, they identified three independent predictors of significant fibrosis: LSM, hemoglobin A1c, and alkaline phosphatase in linear regression analysis. These independent predictors were used to make a risk score for improved accuracy in detecting significant fibrosis.

The research question is highly important and warrants further investigation. The paper is generally well written and structured. Study is well designed, prospective, fibroscan and liver histopathology were done blinded. The use of noninvasive evaluation of liver fibrosis in NAFLD is increasing and has several advantages. Until now “optimal” cutoff value for advanced fibrosis in NAFLD has been 8.2kPa according to Eddoves et al 2019 (Gastroenterology) in larger patient cohort.  

Prevalence of advanced fibrosis in this study was low 11.4%. Was power calculation done for the study?

Higher BMI and CAP values were associated with false positive findings in LSM. Is this due to longer distance between liver capsule and skin?  Authors should discuss mechanical limitation (if any) of LSM in obese subjects. Did the authors find any cutoff value of distance between liver capsule and skin to diminish the number of false positives (LSM>12,8kpa and no advanced fibrosis)?

The liver biopsies were graded blinded for steatohepatitis. Did the authors find any correlation between higher LSM values and steatohepatis present in the histopathology? Weiss and his colleagues (2016 Scand Journal of gastroenterology) documented higher LSM values in NASH group. This should be discussed.

The predictive model FibRO-3 should be validated in independent cohort before wider use as authors suggest.

Reviewer 3 Report

01rst February 2022

Manuscript ID: JCM-1579453

Title: The utility and diagnostic accuracy of transient elastography in 2 adults with morbid obesity: A prospective study.

Journal: JCM

Comments to the Authors:

The authors have elaborated an interesting article about the diagnostic accuracy of Fibroscan in predicting significant liver fibrosis (fibrosis stage ≥ 2) in morbidly obese patients (BMI ≥ 40 kg/m2) with nonalcoholic fatty liver disease 29 (NAFLD). Authors found that Fibroscan utility in predicting significant liver fibrosis in morbidly obese subjects with NAFLD is limited with accuracy of 71.3%. Moreover, a model incorporating hemoglobin A1c and alkaline 45 phosphatase with LSM improved accuracy in detecting significant fibrosis in this patient population.

It is a very interesting and well-elaborated article. By the way, the manuscript needs some commentaries.

Minor points:

  • As the authors said, the lower prevalence of fibrosis stage ≥ 2 is a limitation.
  • Another one is that the proposed predictive model needs to be validated by an independent cohort.
  • There are some spelling errors.
  • Revise References.

Reviewer 4 Report

The authors present a prospective study evaluating the diagnostic accuracy of Fibroscan in predicting significant liver fibrosis (considered as fibrosis stage ≥ 2) in patients with BMI ≥ 40 kg/m2 undergoing bariatric surgery. Overall, the study is original, well-designed and clearly presented by the authors. However, there are some concerns that have to be addressed. Point-by-point:

1. Abstract:

  • there is no a brief introduction to the problem to be addressed in the study
  • the study evaluates the diagnostic accuracy of Fibroscan in morbidly obese patients but not necessarily with NAFLD (there is a % of patients without NAFLD in the study cohort) 
  • how the predictive model was developed is not completely clear in the abstract 
  • I would suggest to mention that LSM is associated with significant fibrosis before the cut-off 

2. Introduction:

  • The scientific rationale is very complete but a hypothesis is missing.
  • The first part of the last paragraph "The present prospective investigation... to fibrosis stage 2 or higher.(17)" seems repetitive and should be better mentioned in other sections such as methods and discussion. 

3. Methods:

  • Given that the "IRB" abbreviation has not been mentioned previously, I suggest to remove it.
  • Please, explain with more detailed which criteria has been used to stablished NASH and fibrosis by liver biopsy. 
  • In the statistical analysis it is not mentioned wether the distribution of normality of the variables has been performed before using mean and SD. Why Wilcoxon test has been used instead of students-t test? Have you considered multicollinearity of variables in the multivariable regression analysis? 
  • A sample size is missing, please provide.

4. Results:

  • A final cohort of 167 patients was included. How many patients were evaluated and excluded? 
  • Transient elastography has been mentioned above and abbreviated to TE. Please, unify all the abbreviations. 
  • As far as I understand, 30% of the patients did not have NAFLD. Please, correct all the statements mentioning that your population was obese patients with NAFLD (not all were NALFD).
  • In addition to % and p-values, I'm missing some measures of association such as OR or RR. Can you provide it? even if it is only in the most relevant variables?
  • Some % are not consistent with absolute values in Table 1, please doble check. 
  • Table 3 shows a success rate of 58.8%, however upper limit of the range is higher than 100%, can you explain this result?
  • The analysis of risk factors of false positive results is very interesting. To evaluate whether there is an increasing risk of getting a false positive result with increasing values of BMI as a continuous variables would be very interesting. I'm also wondering whether you evaluated other clinical/laboratory variables such as having diabetes or high levels of TG.
  • The total number % of false + and true - is not 100%. Please, doble check and correct it.
  • I don't understand how you calculated the predictive values. According to the manuscript, LSM has a PPV of 24.6% and a NPV of 95.5% in discriminating significant fibrosis. However, when you calculate PPV from true positive cases/positive cases, the % is 74%. The same for the NPV which is 71% from the data provided in Table 4. Please, revise this and the S, E and accuracy as well as for the model. 
  • You compare the new model with other models reported in the literature, however, no p-values to affirm that there are significant differences are provided. Can you please add it?
  • A p-value of figure 2 would also be interesting.

5. Discussion:

  • The discussion addresses the important results of the study. Please, adapt it according to the modifications made in the methods/results.

Reviewer 5 Report

I have read with great interest the manuscript proposed by Ali et al. entitled "The utility and diagnostic accuracy of transient elastography in adults with morbid obesity: A prospective study."

Before further processing the authors must address some issues:

  • Lines 67 - 80 : The authors report the characteristics of TE. I believe it is advisable to report also the drawbacks of this elastography techniques, especially in those patients with high BMI (cite and refer to: 10.23736/S2724-5985.20.02773-7)
  • Lines 83-107: The authors focus on the potential role of BMI as confounding factors. However, is it the impact of BMI or abdominal wall (in terms of skin-to-liver distance) the more influent factor? (cite and refer to: 10.3390/diagnostics10100795)
  • Lines 172-176: what test was used to study the normal distribution of variables?
  • Lines 200-224: the authors describe model development, however the authors did report only the discriminative ability of the model (through AUROCS analyses) and not the model calibration (cite and refer to: 10.3390/diagnostics10090619)

Round 2

Reviewer 2 Report

The authors have provided a nicely detailed and thorough response to the comments from the previous review and have addressed my major concerns.

Reviewer 4 Report

The study is very interesting and all comments have been duly addressed. My only concern is that the hypothesis is not clear and is usually formulated before the objectives that will try to confirm/disprove it. 

Reviewer 5 Report

The authors have plentifully addressed reviewers' concerns.